
# Review article:
# Interdisciplinary Perspectives on Climate Sciences – Highlighting Past and Current Scientific Achievements

Vera Melinda Galfi[1], Tommaso Alberti[2], Lesley De Cruz[3,4], Christian L. E. Franzke[5,6], and Valerio Lembo[7]

[1]Institute for Environmental Studies, Vrije Universiteit Amsterdam, Amsterdam, the Netherlands
[2]Istituto Nazionale di Geofisica e Vulcanologia, Rome, Italy
[3]Royal Meteorological Institute of Belgium, Brussels, Belgium
[4]Vrije Universiteit Brussel, Brussels, Belgium
[5]Center for Climate Physics, Institute for Basic Science, Busan, Republic of Korea
[6]Pusan National University, Busan, Republic of Korea
[7]Institute of Atmospheric Sciences and Climate, National Research Council of Italy (CNR-ISAC), Bologna, Italy

**Correspondence:** Vera Melinda Galfi (v.m.galfi@vu.nl)

**Abstract.** In the online seminar series "Perspectives on Climate Sciences: from Historical Developments to Future Frontiers", which took place during 2020-2021, well-known and established scientists from several fields – including mathematics, physics, climate science and ecology – presented their perspectives on the evolution of climate science and on relevant scientific concepts. This special issue aims to create a platform for a more detailed elaboration of the topics discussed in the seminars

but also to publish new scientific findings. In this paper, we first give an overview of the content of the seminar series, and then we introduce the written contributions to this special issue. In line with the spirit of the seminar series, this paper is structured along thematic areas of the broad field of climate science, conveying different perspectives on the climate system: geophysical fluid dynamics, dynamical systems theory, multiscale processes, statistical physics, paleoclimate and the human dimension.

## 1 Introduction

The "Perspectives on Climate Sciences: from Historical Developments to Future Frontiers" online seminar series (POCS, 2021) took place from September 2020 to July 2021, in the middle of the COVID-19 pandemic. The seminar series materialised from our belief that it is important for Early Career Scientists (ECS) to learn more about their research field's broader scientific background and historical context. Positioning their own research in the "big picture" gives them additional motivation to pursue their scientific goals, and shows them possible scenarios for the future evolution of climate sciences. Especially engaging

for young researchers is to learn about the history of their research field directly from the pioneers who actively shaped the path of scientific progress. With this intention in mind, we invited well-known and well-established scientists to speak in front of a broad audience, including, but not restricted to, ECSs. They were asked to talk about their contribution to and perspective on the development of climate sciences, both from a scientific and personal perspective, including the challenges they faced as young scientists. It was a great opportunity to get an insight into the scientific career paths of the speakers that were often far from





straightforward. Their careers were hindered by many unforeseen difficulties, even self-doubt, but at the same time pushed forward by perseverance, dedication to science and sometimes a bit of luck. This gave more self-confidence and personal motivation to many of us, especially ECSs, which was sorely needed in times of isolation due to the COVID-19 lockdowns and geopolitical turmoil.

Climate science emerges at the interface of several disciplines, such as physics, mathematics, chemistry, biology, and the
sphere of social sciences. Hence, it is highly interdisciplinary. A further aim of the seminar series was to stimulate the interaction among these disciplines. In David Ruelle's seminar, we learned that, in his opinion, the most productive period in terms of interdisciplinary work took place between 1970 and 2000, and is connected to the development of chaos theory, building upon seminal works of mathematicians, physicists and meteorologists, like Henri Poincaré and Ed Lorenz. After this period, each discipline developed mainly on its own path, drifting apart from each other. Nowadays, we are facing what is probably
the biggest challenge of humankind: the ongoing global climate crisis. In order to be able to address this existential crisis, we need to exchange ideas and methods among the above disciplines and enhance interdisciplinary interactions. The successes achieved in the 1990s in closing the Ozone hole in the Arctic stratosphere, as outlined in Susan Solomon's seminar, give us confidence that this is indeed possible and achievable.

The recognition of the well-posedness and rigorous mathematical background of the physics behind climate science was
formalized in 2021, when the Nobel Prize in Physics was awarded to Klaus Hasselmann, Syukuro Manabe and Giorgio Parisi (Franzke et al., 2022). This was a milestone in the acknowledgement of the necessary relation between these disciplines, i.e. the broad cohort of science, technology, engineering, and mathematics (STEM) on one side and human sciences on the other side. In his seminar, Hans von Storch reviewed the issue of post-normal science and basic norms for natural scientists (so-called CUDOs referring to Communalism, Universalism, Disinterestedness and Organised scepticism), while Klaus Hasselmann ad-
vocated in his interview the social role of the climate scientist. Eugenia Kalnay updated us on the newest developments in terms of coupling between Earth System Models (ESMs) and Human System Models (HSMs). Altogether, these precious contributions emphasised the uncomfortable but challenging situation in which the climate scientist (meant in the broadest possible sense) is required to engage in the interaction with stakeholders and policymakers, within and beyond the longstanding format of the Intergovernmental Panel for Climate Change (IPCC), jointly established by the World Meteorological Organization
(WMO) and the United Nations Environment Programme (UNEP) in 1988.

The aim of this special issue is to create a platform for a more detailed elaboration of the topics discussed in the seminars, but also to publish new scientific findings. In this paper, we first give an overview of the content of the seminar series in Section 2, and then we introduce the written contributions to this special issue in Section 3.

## 2 The "Perspectives on climate sciences" seminar series

The interdisciplinary background of the speakers revealed several different perspectives on the climate system and climate science. In this section, we present shortly the content of the seminars in line with these perspectives. The *Geophysical Fluid Dynamics (GFD)* perspective is a fundamental one. It provides the mathematical equations describing the evolution of the





climate system and its main components, and thus the basis for weather and climate modelling. The *dynamical systems* per-
spective focuses on the evolution of these equations - or rather of their reduced-order versions - in the phase space, using

mathematical tools to analyse properties of different states of the system, such as stability and predictability. The *multiscale*
perspective complements the dynamical systems view, interpreting the weather and climate as a high-dimensional complex
system exhibiting scale-invariance. When modelling the climate system, a *stochastic* behaviour emerges due to the sensitive
dependence on the initial conditions and the impossibility of resolving the processes down to the scale of viscosity. Thus, the
stochastic perspective is indispensable for accurate weather predictions and a proper interpretation of climate model output.

The *human dimension* introduces additional uncertainty to projecting the future evolution of the system, highlighting at the
same time that humans are an essential system component.

## 2.1 The climate system from the perspective of geophysical fluid dynamics

In order to understand the complex dynamics of the climate system beyond the radiation balance, and disentangle internal
variability from external forcing, one needs to model the time evolution of its components and their interactions. These include

the hydrosphere, cryosphere, lithosphere, biosphere and what affects society most on a daily basis: the atmosphere. Geophysical
fluid dynamics (GFD) describes the evolution of these components based on the principles of mechanics and thermodynamics.
These principles can be expressed as partial differential equations, in an appropriate form of the Navier-Stokes equations,
adapted to the relevant processes at these vast scales, such as stratification and the Coriolis effect.

In order to describe the evolution of an immensely complex system such as the atmosphere and gain insight into its dynamics,

the relevant degrees of freedom and conserved quantities need to be identified. Brian Hoskins recounted how the concepts of
(quasi-geostrophic) potential vorticity (Rossby, 1938, 1940) enabled the development of the first numerical weather prediction
models and how isentropic potential vorticity maps helped gain an understanding of weather developments (Hoskins et al.,
1985).

A century before the introduction of these concepts, severe weather phenomena such as hurricanes had already motivated

early meteorologists to analyse their rotating behaviour on weather maps and understand their dynamics. Kerry Emanuel
described how the scientific progress around hurricanes took a step backwards, highlighting a persistent misunderstanding
around the processes driving these dynamics. Despite progress made in the 40s and 50s (Kleinschmidt, 1951; Riehl, 1950),
later research ignored the importance of surface enthalpy fluxes and instead focused on latent heat release. This led to a
failure of numerical simulations of hurricanes and errors in textbook descriptions persisting even until today. Finally, improved

observations played a key role in confirming the dominant drivers.

Raymond Pierrehumbert pointed out in his talk that as simple as the rules governing an atmosphere with water vapour might
be – i.e. Newtonian mechanics, thermodynamics, and radiative transfer – nontrivial properties emerge in the resulting climate
system. Again, misconceptions such as the thermostat hypothesis remind us to stay vigilant and encourage us to try reproducing
earlier work. This hypothesis, which claimed that the temperature in the tropics could not exceed 302K due to shading by high

convective clouds, was debunked with some effort (Pierrehumbert, 1995). In his more recent work, Pierrehumbert (2005)





showed how GFD ideas can be revisited on the new testing ground provided by the rich variety of exoplanetary atmospheres, including tidally-locked planets and atmospheres with supercritical water.

On our own planet, again, Clara Deser explained how climate projections driven by GFD simulations help understand regional climate change effects. On these scales, internal variability has an enormous impact on the observed local trends in atmospheric variables. In her seminar, she presented a solution to robustly disentangle the effects due to the external forcing (anthropogenic, volcanic, etc.), the model differences, and the system's internal stochastic variability: the latter can be identified as the variability observed in large initial-condition ensembles of historical climate simulations (Deser et al., 2012, 2020).

The disentangling of internal and forced variability was also the central theme of Michael Mann's story of the Atlantic Multidecadal Oscillation. This concept emerged based on the multidecadal (50-70y) spectral peak in global surface temperature records, and rapidly gained traction in scientific works, but was subsequently shown to be an artefact caused by the response of the climate system to both anthropogenic and natural forcings, rather than an internal mode of variability. Strong evidence for this was provided by the continued presence of this spectral peak in the globally averaged ensemble mean temperatures of CMIP5 climate control simulations of the last millennium. Indeed, one would expect the internal variability to be averaged over these ensembles, meaning that any remaining variability can only be attributed to external forcings (Mann et al., 2021).

## 2.2 The climate system from the perspective of dynamical systems theory

The atmosphere and the climate are chaotic dynamical systems, given the sensitive dependence of their evolution on initial conditions. David Ruelle discussed in his seminar the interdisciplinary origins of chaos theory. We learned that the mathematician Henri Poincaré (Poincaré, 1908) knew already at the beginning of the 20th century that the weather is chaotic. He stated that meteorologists are not able to predict the exact location of a cyclone because of imperfect and insufficient initial conditions, which makes the weather to be perceived as something random. However, the chaotic nature of the weather did not become generally established until the work of Ed Lorenz (1963), strongly connected to the spread of numerical computers in the middle of the 20th century. As David Ruelle reported, this was followed by an explosion of interdisciplinary work on chaos theory between 1970 and 2000, involving meteorology, hydrodynamics, chemical kinetics and the astronomy of the solar system (Ruelle and Takens, 1971; Li and Yorke, 1975; Laskar, 1989). After that, interdisciplinarity declined and the work nowadays is mainly done in individual disciplines.

In contrast with the GFD perspective, which focuses on equations describing the temporal evolution of dynamic and thermodynamic variables, the dynamical system perspective considers the solution of these equations in the phase space, the set of all possible configurations of the system. In the phase space, one studies trajectories corresponding to different initial conditions, the stability of fixed points, periodic and non-periodic orbits, the geometry of the attractor, and the properties of the invariant measure. Despite the high level of mathematical abstraction, these concepts are useful because they provide unique insights into the system's dynamics. It becomes clear, for example, that the geometrical properties of the attractor are strongly related to the evolution of trajectories, and thus to the dynamics of the system. Similar states or configurations of the climate system are close to each other in the phase space, and evolve corresponding to the local geometry of the attractor in that region, even if they are far away from each other in time. However, as we found out in James Yorke's seminar, the dynamics of a chaotic





system can be very complex, due to phenomena such as hetero-chaos (Saiki et al., 2018, 2021). Because of this, very similar initial conditions can lead to completely different evolutions of the weather, and thus to serious predictability issues.

These predictability issues are well known in numerical weather prediction, where it has become clear over time that purely deterministic predictions are insufficient and misleading. Tim Palmer convinced us in his seminar that the predictability of a nonlinear system and the related uncertainty always depend on the initial condition. Thus, further developments in probabilistic

prediction, based on enough ensemble members, are crucially important for the future of weather prediction. Also important for the future are exploiting the possibilities offered by artificial intelligence as well as using stochasticity for improved parameterisations and data assimilation and for making simulations computationally less expensive.

Phenomena such as climate change, tipping points, and internal variability cannot be explained based on the classical concept of an attractor with an invariant measure. In his seminar, Michael Ghil talked about pullback attractors of non-autonomous

dynamical systems, i.e. with time-dependent forcing (Crauel and Flandoli, 1994; Ghil et al., 2008; Chekroun et al., 2011; Drótos et al., 2015). Already, the presence of noise in the form of internal variability makes the notion of a pullback attractor necessary. In the case of constant external forcing and stochastic natural forcing (internal variability), the corresponding attractor is a time-invariant pullback attractor, referred to as a random attractor. The effect of changing radiative forcing turns the attractor into a time-dependent pullback attractor. Denisse Sciamarella showed us how the study of the attractor's topology, with the help

of branched manifolds, helps us to understand chaos and the effect of noise. Due to noise-induced chaos, the attractor can be described by different homology groups at different time steps (Sciamarella and Mindlin, 2001; Charó et al., 2021).

## 2.3 The climate system from a multiscale perspective

The climate system is a complex system, featuring a large number of degrees of freedom with mutually interacting components in a nonlinear way (Franzke and O'Kane, 2017; Dijkstra, 2013). These concepts are strictly related to those relying on the

description of turbulent flows, as the existence of a hierarchy of exponents to fully characterise the nature of the system (i.e., the so-called multifractal view, Frisch, 1995), the observation of a global vs. a local scale invariance (Kuzzay et al., 2017), and the role of extreme events as singularities (Alberti et al., 2023).

Shaun Lovejoy applied the above concepts to describe different dynamical regimes in the climate system: the weather, the macroweather, and the climate (Lovejoy, 2019). Indeed, weather and climate models are based on thermodynamics and

continuum mechanics and are successful because they retain only some relevant "macroscopic" variables. However, the role of the "details" (cfr. Mandelbrot, 1977) cannot be ignored, requiring the development of a higher-level description of unknown aspects of the climate system. This led to several (high-level) macroweather and climate models based on energy balance and scale invariance, which profoundly affect our forecasting capabilities at monthly and seasonal scales, as well as improved multidecadal climate projections.

Berengere Dubrulle went deeper into the "details" by focusing on a ubiquitous aspect of fluids: turbulence (Dubrulle, 2022). Turbulent flows are characterized by a self-similar energy spectrum, the signature of fluid movements at all scales. This organization has been described for more than 70 years by the phenomenology of "Kolmogorov/Richardson cascade"(Kolmogorov, 1941; Richardson, 1922): the energy injected on a large scale by the work of the force that moves the fluid (e.g., a turbine) is





transferred to smaller and smaller scales with a constant dissipation rate, up to the Kolmogorov scale, where it is transformed
into heat and dissipated by viscosity. Such cascade phenomenology is at the basis of most turbulent models. She discussed how
progress in numerical simulations and laboratory experiments gradually changed such simple vision (starting from Landau's
objection in the 50's), leading to a new picture where quasi-singularities living beyond the Kolmogorov scale play a central
role (Dubrulle et al., 2022). This has important impacts on resolution requirements of numerical simulations and calls for new
models of turbulence (Dubrulle and Gibbon, 2022).

Ken Golden focused instead on a key component of Earth's climate system: the polar sea ice. Indeed, sea ice exhibits com-
plexity on length scales ranging from tenths of millimetres to tens of kilometres. A principal challenge in modelling sea ice
and its role in climate is how to use information about the small-scale structure to find the effective or homogenized properties
on larger scales relevant to coarse-grained climate models. In other words, and strictly connected to previous concepts intro-
duced by Shaun Lovejoy and Berengere Dubrulle, how can we predict macroscopic behaviour from microscopic laws? Similar
questions arise in statistical mechanics, materials science, and many other areas of science and engineering (Golden et al.,
2020b). In his talk, he gave an overview of recent results, inspired by composite materials and statistical physics theories, on
modelling effective behaviour in the sea ice system over a broad range of scales (Golden et al., 2020a). He also discussed how
physical sea ice processes influence microbial communities in the ice and upper ocean and vice versa. This work is helping to
advance how sea ice is represented in climate models, and to improve projections of the fate of the Earth's sea ice packs and
the ecosystems they support.

## 2.4 The climate system from the perspective of stochastic processes and statistical physics

When modelling the climate system, a *stochastic* behaviour emerges due to the sensitive dependence on the initial condi-
tions and the impossibility of resolving the processes down to viscosity. This makes the stochastic perspective indispensable
for accurate weather predictions and a proper interpretation of climate model outputs. At the same time, stochastic methods
are useful for different aspects of weather and climate modelling, such as parametrisations, to model certain behaviours and
processes in the climate system and even to reduce computational costs. Due to its focus on many-particle systems and the
expertise in connecting micro- and macroscales, statistical physics offers inspiration and practical tools for understanding the
climate system.

As a format break from the other seminars, the authors interviewed Klaus Hasselmann, who won the Nobel Prize in Physics
in 2021 (Franzke et al., 2022; Bohémier, 2022) together with Syukuro Manabe and Giorgio Parisi. Klaus Hasselmann pioneered
the development of stochastic climate models (Hasselmann, 1976) and reduced order models (Hasselmann, 1988) in order to
detect climate change signals (Hasselmann, 1993). During the discussion, Klaus Hasselmann stressed the importance of his
background in physics for his scientific career, how he opened up new research fields and jumped to new endeavours when he
got excited by new ideas, without ever losing curiosity and fun doing research. His work on stochastic climate models led to
the development of stochastic climate theory and stochastic parametrisations, now in use in operational models.

Cecile Penland used Klaus Hasselmann's model reduction approach to develop Linear Inverse Models (Penland, 1989).
Linear Inverse Models are linear dynamical models driven by stochastic noise. In her presentation, she described her scientific





path and how she started using stochastic methods. She emphasized the concept of probability as a physical, conserved quantity, and pointed out the necessity to use this concept more often in climate science. However, she also stressed that stochastic
methods have become more widespread nowadays.

Roberto Benzi introduced the concept of stochastic resonance to explain past climate variability, together with Giorgio Parisi (Benzi et al., 1982). In his presentation, he outlined the many difficulties and misconceptions he faced when first introducing the concept, both in the fields of climate dynamics and nonlinear dynamical systems. In the meantime, stochastic resonance has been used in a large number of physical systems.

In his seminar, Giovanni Jona-Lasinio helped us understand the state of non-equilibrium based on macroscopic fluctuations theory. He argued that non-equilibrium is characterised by a variety of phenomena, thus searching for a unique theory, as the one describing classical thermodynamics, is prone to fail, and one has to restrict to subclasses of problems. He pointed out that one of the major difficulties is defining adequate thermodynamic functionals in states far from equilibrium, and showed how large deviation rates are useful in this regard. Large deviation theory is currently applied in climate studies (Gálfi et al., 2021).

## 2.5   A paleoclimate perspective on the climate system

Paleoclimate broadens our perspective on the climate system due to the very long time scales considered. These long time scales are essential to understand the dynamics of the whole system, including the effect of the slow components, such as the hydrosphere and the biosphere.

Pascale Barconnot emphasised in her talk that knowledge about past climate states brings us closer to understanding future
changes and helps us evaluate climate models. Based on paleoclimate modelling, climate models can be tested under different conditions than we experience today. One can test the effect of using different sets of parameters, applying different forcings, or conducting simulations based on different experimental protocols. All this is important in order to be confident that models can indeed simulate the climate of the future. The paleoclimate perspective reveals that the various evolution periods in the history of the climate system, do not just have a distinct mean state, but are instead the result of very different system dynamics,
with different internal variability and feedbacks.

Valerie Trouet pointed out, based on tree ring studies, the strong interconnection between different climate system components, discussing the remote effects of the atmospheric dynamics in the upper troposphere, on the vegetation growth and wildfires at the surface. Furthermore, she emphasised the devastating effects of fire suppression on the frequency and extension of Californian wildfires. Disturbances are often part of the natural dynamics of the system, and altering their natural occurrence
frequency can lead to the opposite of the intended outcome.

## 2.6   The human dimension of the climate system

On the one hand, humans are an essential part of the climate system. Due to the emission of greenhouse gases, land use, and other human activities disturbing the natural balance, they directly affect the state of the system. However, at the same time, the state of the climate influences human activities and societies. On the other hand, the human dimension represents a




huge challenge for modelling the future development of the climate system due to the difficulty – or even impossibility – of describing human behaviour and the evolution of societies based on mathematical equations.

Building upon Hasselmann's seminal ideas about climate response and attribution, Gabriele Hegerl gave a retrospective survey of how ideas such as optimal fingerprinting, detection and attribution of climate change were brought up in years by her and her colleagues, mentioning Ben Santer and Klaus Hasselmann among others. The rationale of identifying causes of climate change was not only limited by the emergence of an anthropogenic signal, but also by the issue of separating forcings and feedbacks in order to address the feasibility of an "orthogonal" approach to the attribution of change to specific forcings, e.g. greenhouse gases or aerosols. Spatial patterns of the expected forced response (after Hasselmann) were in this respect compared to observations, in order to isolate the role of internal variability. New perspectives are currently being drawn in the field by starting to apply the ideas of attribution to the investigation of the impacts of climate change.

In a thought-provoking talk, Hans von Storch stressed the evolution of the study of climate through the 20th and first part of the 21st Century, starting from the concept of "climate determinism", aimed at justifying the European colonisation of Africa and Asia, towards a proper rigorous method-based science, relying on CUDOS principles, towards the issues of postnormal sciences. Von Storch suggested that the scientific process has been challenged by the increasing demand for science-motivated policies. At the same time, policymakers had been progressively asked to take decisions aimed at addressing climate change adaptation and mitigation. In this context, the climate scientist has been required to take a political stance, and motivating public opinion has been, at times, given higher priority than producing additional scientific results. Von Storch then gave tips on how climate scientists may navigate these troubled waters and what are, in his opinion, the prospects ahead.

The contributions by Susan Solomon and Eugenia Kalnay went deeper into how much the interaction between policymakers and climate scientists is relevant in order to address the challenges posed by the changing climate. Susan Solomon gave an insightful retrospective on the path to the Montreal agreement, which led to a worldwide ban on emissions responsible for widening the ozone hole over Antarctica. This is regarded as a pioneering step on the virtuous interaction between policymakers and scientists, with scientific evidence motivating the choice of the community of nations. This step has paved the way to the establishment of the United Nations-led IPCC, the recurrent redaction of the Assessment Reports and the organisation of the Conference of Parties (COP).

Eugenia Kalnay's contribution focused on stressing the importance and urgency of integrating the human system into Earth System modelling. This is not only related to an innovative approach to the redaction of socio-economic development pathways, serving as boundary conditions to Earth system model exercises. The human system, according to Kalnay, has to be viewed as a module and to be integrated into the Earth system model via coupling to the other components, such as the atmosphere, hydrosphere, and cryosphere. The task is becoming more urgent as most of the world's population is increasingly reliant on fossil fuel combustion, urbanization, and the continuous increase in Gross Domestic Product. Unlike the usual approach in the climate model communities, the Earth system-human system coupling has to be bidirectional, and target the specific subdomains. According to Kalnay, the main challenge embedded in this viewpoint is addressing socio-economic policies, especially those that involve the modification of the current social inequalities.



## 3   This special issue

The contributions to this special issue emerged either from the motivation to elaborate with more details or to complement the topics discussed during the seminars, which resulted in a collection of both research and review articles and one brief communication.

A central problem of climate projections, driven by *geophysical fluid dynamics* simulations, is how to disentangle the internal variability of the climate system from the forced response. The contribution by Clara Deser (Deser and Phillips, 2023) involves the theoretical reasoning behind the necessity to implement large ensemble model simulations in order to solve this problem. The focus is on three main aspects: producing large ensembles with different initial conditions, the role of observations and the technique of dynamical adjustment to remove the effect of the general circulation from observed records and to obtain a more reliable estimate of the forced response. As a case study, the recent trends in European climate are investigated.

Kalnay et al. (2023) focus on a the importance of considering all components in the climate system for numerical weather prediction and point to the advantages of strongly coupled data assimilation improved by machine learning, using models of different complexity. In particular, they start from a simple Lorenz model by detecting slow and fast modes of motion. Variational and ensemble coupled data assimilation methods are compared in a coupled quasi-geostrophic model (De Cruz et al., 2016), and the ensemble Kalman filter is chosen for comparing weakly and strongly coupled data assimilation in general circulation models of intermediate complexity. Finally, neural networks are employed to improve the effectiveness of correlation-cutoff techniques in addressing data assimilation with small-size ensembles.

Michael Ghil and Denisse Sciamarella (Ghil and Sciamarella, 2023) discuss the relationship between climate science and *dynamical systems theory*. The theoretical concepts and methods have influenced climate science since the 1960s. Recently, increased developments benefited from advancements in computing resources and observational capabilities. The text further explores the contributions of nonlinear dynamics and the theory of non-autonomous and random dynamical systems to understanding the interplay between natural variability and anthropogenic climate change, as well as the role of algebraic topology in shedding light on this relationship. The review concludes by addressing tipping points and transitions in climatic behaviour under time-dependent forcing.

Dorrington and Palmer (2023) also take the dynamical systems perspective and study the interaction between stochastic forcing and regime dynamics in a barotropic $\beta$-plane model. They discuss a rather counter-intuitive result showing that stochastic forcing can increase the low-frequency variability of the system and thus the persistence of certain regimes. This example motivates us to better understand the impact of stochastic physics on regimes in full-blown climate models.

Shaun Lovejoy (Lovejoy, 2023) interprets the atmosphere from a *multiscale* perspective and reviews the development of scaling notions in atmospheric sciences. Until the 1980s, scaling was limited to self-similar cases. However, recent developments in multifractals and generalized scale invariance allowed for characterising and modelling strongly intermittent scaling processes and anisotropic systems. These generalisations are crucial for atmospheric applications. Scaling is now considered a general symmetry principle that helps define dynamical regimes in weather, macroweather, macroclimate, and megaclimate. Anisotropic scaling systems, such as atmospheric stratification, require new definitions of scale to understand structures' mor-





phologies systematically. The text also addresses challenges in widely accepting the scaling paradigm and its implications for alternative scaling approaches in weather and climate models, including long-range forecasts and climate projections.

Golden et al. (2023) focus on a specific but very important problem of climate modelling, the right representation of the polar sea ice in climate models. Sea ice is a multiscale composite material, exhibiting complexity on length scales ranging from tenths of millimetres to tens of kilometres. They show how to model the role of sea ice in the climate system by connecting microscales and macroscales based on Stieltjes functions.

The contribution of Jona-Lasinio (2023) is related to the very important though the challenging task of connecting micro- and
macroscales as well, but from the perspective of *statistical physics*. The author gives an overview of macroscopic fluctuation theory, discussing large deviation rates in non-equilibrium physical systems, the differences between equilibrium and non-equilibrium states, and the applicability of this theory to the climate system.

*The human dimension* takes a central role in the brief communication by Hans von Storch (von Storch, 2023). This contribution is aimed at giving a wide overview of the social implications of climate sciences through recent history, from climate
determinism giving a scientific background to western colonisation, to the ideas of post-normal science and basic scientific norms, the so-called CUDOS: Communalism, Universalism, Disinterestedness and Organised Skepticism.

## 4  Outlook

The online seminar series gave insight into the history of major methodological developments to better understand and simulate the complex climate system. It also highlighted the importance of fundamental mathematical and physical science for making
progress in understanding the climate system, which was recently prominently illustrated by the award of the 2021 Physics Nobel Prize to the climate scientists Syukuro Manabe and Klaus Hasselmann, and the physicist Giorgio Parisi (Franzke et al., 2022). Especially Klaus Hasselmann and Giorgio Parisi used fundamental mathematical methods to understand climate change. The speakers in the seminar series and their contributions to this special issue also demonstrate that even today further fundamental methodological mathematical and physical advances are needed to improve our understanding of the complex climate
system. Dealing with the additional complexity, which arises by integrating human behaviour as an active part of the physical system, requires new groundbreaking tools and concepts. We hope that the seminar series and this special issue will motivate such further developments.

*Data availability.*  The presentation slides of the seminars are available at: https://sites.google.com/view/perspectivesonclimate/materials.

*Author contributions.*  All authors contributed to the writing of the manuscript.



*Competing interests.* At least one of the (co-)authors is a member of the editorial board of Nonlinear Processes in Geophysics.

*Acknowledgements.* CF is supported by the Institute for Basic Science (IBS), Republic of Korea, under IBS-R028-D1. LDC acknowledges support from the Belgian Science Policy Office (BELSPO) through the FED-tWIN programme (Prf-2020-017). VMG acknowledges the support of the Water and Climate Risk Department, Institute of Environmental Studies, Vrije Universiteit Amsterdam.





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
