# Peer review of "Review article: # Interdisciplinary Perspectives on Climate Sciences – Highlighting Past and Current Scientific Achievements"

_Nonlinear Processes in Geophysics, 2023_

## Referee Comment (RC1)

This is an introductory paper to a Special Issue of *Nonlinear Processes in Geophysics*, which is itself the outcome of a series of seminars organized by the authors.

The seminars were intended at giving a large overview of the present state of climate sciences and of the associated perspectives. The seminars were very successful and a number of contributors have written articles for the present Special Issue, which was edited by the organizers of the seminars (and authors of the present paper). Some contributors went beyond the content of their original seminars, writing in particular broad review papers.

One major feature of climate sciences is that they bear on an extremely wide range of different disciplines, going for instance from the theory of dynamical systems (in the development of which the study of the atmosphere played a critical role) to stochastic physics and even social sciences. The organizers have been able, both in the seminar series and in the present Special Issue, to gather contributions which cover much of that wide range of climate sciences.

The present introductory paper gives an overview of the original seminar series, followed by a more detailed description of the eight papers included in the Special Issue. It is succinct but very clearly written. It is actually not only an introductory paper to the Special Issue, but it can certainly be used, by a scientifically educated but not specialist reader, as a first general introduction to climate sciences.

I have a very favorable opinion of this paper, and strongly recommend its acceptance. I congratulate the authors for the work they have done for setting up and organizing the seminar series, setting up and editing the NPG Special Issue, and finally writing the present introductory paper. I have no doubt that, as they hope, the seminar series and the ensuing Special Issue will motivate further developments.

I give below a number of editing suggestions

1.  Ll. 58 and 173, I suggest … *the **practical** impossibility of …*

L. 101-102, I do not think the climate (if defined as the whole attractor of the system) can be said to be chaotic. I suggest *The atmosphere is a chaotic dynamical system, given its sensitive dependence of its evolution on initial conditions.*

L. 103, … *Henri Poincaré* […] *knew already* […] *that the weather is chaotic*. I would rather suggest *Henri Poincaré* […] *had already explicitly considered* […] *the possibility that the weather is sensitive to initial conditions.*

L. 106, I would suggest … *until the work of Ed Lorenz (1963) and the works that followed, all strongly connected* …

Ll. 173-174, … *the stochastic perspective indispensable for accurate weather predictions* … I think it is more a question of usefulness of the weather predictions than of strict accuracy.

L. 175, The word *parametrisations* has not been defined at this stage. I would suggest for instance *parametrisations (i.e. representation of the impact of structures that are not resolved by the numerical model onto structures that are resolved)*

L. 98, expand CMIP (*Coupled Model Intercomparison Project*) and give reference

L. 106, paper Lorenz (1963) is missing from the list of references

---

## Editor Comment (EC1)

The two referees of the paper have now submitted their evaluations. The first referee recommends acceptance of the paper, subject to a number of minor editing corrections. The second one (who has let his name known, and is Valerio Lucarini) recommends acceptance without further modification of the paper.

I intend as Editor to follow the referees' advice. But, before I can take a final decision, I must wait for the authors' final response. They have been requested to send it by 21 Feb 2024.

They should explain how they have dealt with the suggestions of the first referee.

And for a final remark, the proper (accented) spelling of names are as follows

- Ll. 150 and 164 (and maybe elsewhere, please check). Berangere $\rightarrow$ Bérangère (two accents, one acute, one grave)

- L. 186, Cecile $\rightarrow$ Cécile

- L. 211, Valerie $\rightarrow$ Valérie

I would suggest to make all these corrections. Now, from what I can see at link https://sites.google.com/view/perspectivesonclimate/materials, only Cécile Penland seems to have put there an accent on her name. So, I leave it to the authors to make the corrections or not.

I look forward to receiving the final version of the paper.

---

## Author Comment (AC1)

Dear Editor,

we are thankful for the interest in our paper and for the very positive evaluation. As listed below, we have addressed every comment and suggestion of Reviewer #1. We have also implemented your spelling suggestions.

Best wishes,
Vera Melinda Galfi (for all authors)

**Reviewer #1**

We would like to thank the Reviewer for their very positive review of our paper and for their editing suggestions, which have helped to polish our text. We detail the changes that have been made to the manuscript below.

1.  *Ll. 58 and 173, I suggest ... the practical impossibility of ...*

We have updated these as suggested.

*L. 101-102, I do not think the climate (if defined as the whole attractor of the system) can be said to be chaotic. I suggest The atmosphere is a chaotic dynamical system, given its sensitive dependence of its evolution on initial conditions.*

We thank the reviewer for pointing this out. We refer here to the dynamics of the climate system itself, i.e. the coupled system including the atmosphere, hydrosphere, biosphere, lithosphere and cryosphere, which does exhibit sensitive dependence on the initial conditions (Ghil and Lucarini, 2020). We do not use the word "climate" in its conventional sense, i.e. referring to the statistics over a period of several decades of the climate system. We express this clearer in the new version of the manuscript.

*2 L. 103, ... Henri Poincaré [...] knew already [...] that the weather is chaotic. I would rather suggest Henri Poincaré [...] had already explicitly considered [...] the possibility that the weather is sensitive to initial conditions.*

We have updated these as suggested.

*L. 106, I would suggest ... until the work of Ed Lorenz (1963) and the works that followed, all strongly connected ...*

We have updated these as suggested.

*Ll. 173-174, ... the stochastic perspective indispensable for accurate weather predictions ... I think it is more a question of usefulness of the weather predictions than of strict accuracy.*

We agree with the reviewer and have changed this sentence accordingly.

*L. 175, The word parametrisations has not been defined at this stage. I would suggest for instance parametrisations (i.e. representation of the impact of structures that are not resolved by the numerical model onto structures that are resolved)*

We have implemented these as suggested.

*L. 98, expand CMIP (Coupled Model Intercomparison Project) and give reference*

We have implemented these as suggested.

*L. 106, paper Lorenz (1963) is missing from the list of references.*

We have added the corresponding reference.

**Reviewer #2**

We would like to thank the Reviewer for their interest and very positive review of our paper.

**References**

M. Ghil and V. Lucarini. The physics of climate variability and climate change. *Rev. Mod. Phys.*, 92:035002, Jul 2020. https://doi.org/10.1103/RevModPhys.92.035002.